# Establishment of a Monitoring Model for the Cotton Leaf Area Index Based on the Canopy Reflectance Spectrum

Xianglong Fan [1], Xin Lv [1,*], Pan Gao [2], Lifu Zhang [3], Ze Zhang [1], Qiang Zhang [1], Yiru Ma [1], Xiang Yi [1], Caixia Yin [1] and Lulu Ma [1]

1   Xinjiang Production and Construction Crops Oasis Eco-Agriculture Key Laboratory, Shihezi University College of Agriculture, Shihezi 832003, China
2   College of Information Science and Technology, Shihezi University, Shihezi 832003, China
3   Aerospace Information Research Institute, Chinese Academy of Sciences, Beijing 100101, China
*   Correspondence: luxin@shzu.edu.cn

**Abstract:** Cotton is the main economic crop in China and is important owing to its use as an industrial raw material and a cash crop. This experiment was conducted in the main cotton-producing area of Xinjiang, China. A hyperspectrometer was used to monitor the canopy spectral reflectance of cotton at different stages of growth. The results showed that the leaf area index (LAI) increased with the increase in the amount of nitrogen fertilizer added during the early full boll stage and decreased with the increase in nitrogen fertilization in the full and late boll stages. Insufficient or excessive fertilization led to a decrease in the LAI. The visible light band indicated that the canopy spectral reflectance decreased, and the amount of fertilizer increased in all the growth stages. The near-infrared band revealed that the canopy spectral reflectance increased with the amount of nitrogen applied during the bud stage, early boll stage, and the most vigorous period of boll growth. During the flowering period, the spectral reflectance followed the order N3 > N4 > N2 > N1 > N0. During the entire growth period of cotton, the values of the cotton LAI predicted using the ratio vegetation index (RVI) model were found to best fit the measured values. The LAI monitoring models of cotton in each growth stage were different. The TVI model is the best in the bud and early boll stages. The NDVI model is the best in the flowering stage, and the DVI model is the best in the full boll stage. This study provides a basis to accurately monitor the LAI in each growth period of cotton.

**Keywords:** cotton canopy; leaf area index; growth period; sensitive band; monitoring model





## 1. Introduction

The leaf area index (LAI) refers to the one-sided green leaf area per unit area of land [1] and can be used to infer information, such as whether the canopy structure is reasonable and whether the vegetative and reproductive growth are coordinated, as well as provide information about the growth process. For example, the LAI has been found to significantly correlate with crop yield [2–5]. Additionally, it is an important indicator of the characteristics of cotton population, as well as the growth of crop population, and the initial energy exchange at the canopy surface [6,7]. However, the accurate estimation of crop LAI in real time has always been a difficult problem in monitoring crop conditions, managing production, and estimating yields. Traditional methods to measure the LAI are primarily based on point data measurement, which cannot reflect the spatiotemporal characteristics of the crop LAI. Moreover, these methods are time-consuming, expensive, and prone to large errors, which reduce the significance of estimating cotton yields. In recent years, with the rapid development of remote sensing technology, this approach has become a new way to monitor the crop LAI [8]. Hyperspectral remote sensing can be used to conduct a direct quantitative analysis of weak spectral differences on ground objects and has been shown to offer strong advantages in estimating the amount, coverage, and biochemical parameters of crops [9–11].

Currently, studies have shown that the plant canopy spectrum is closely related to the LAI [12]. The plant canopy spectrum was found to effectively reflect the changes in LAI when the characteristic parameters of the crop biophysics spectrum were studied [13]. A correlation analysis between the cotton LAI and a single band was primarily concentrated in the visible and near-red bands [14]. Spectral transformation is used to enhance the spectral information of the cotton LAI [15–17]. Although the spectral bands of the cotton canopy can reflect the changes in the cotton LAI, the analysis of a single spectral band will not utilize the spectral information of the LAI in the other bands, thus, reducing the sensitive spectral information of the LAI over the entire band range. However, in recent years, researchers have begun to combine multiple sensitive bands in a mathematical manner to construct a spectral index to more effectively reflect the growth status of vegetation to monitor the changes in the cotton LAI more accurately [18–20]. The spectral index can more fully reflect the spectral information of the cotton LAI, explore the mechanism of response of the spectral index and cotton LAI, and further improve the monitoring accuracy of the cotton LAI. Previous studies have found that many types of models can be used to monitor the LAI spectral index in cotton [21–24], and it is convenient and simple to use the spectral index to establish models to estimate the LAI of cotton.

In conclusion, most studies have applied the sensitive band and the established model to the monitoring of vegetation parameters during the whole growth period. Given that cotton is in a dynamic growth process, the leaf structure, cell shape, and nutrient content required in each period are different, and when the vegetation coverage reaches a certain range, it will not change. However, the LAI may still be increasing, so it is difficult to distinguish the change rule of LAI in different growth periods [25]. Solely determining the sensitive band and modeling the entire growth period cannot accurately reflect the characteristics of LAI in different growth periods [26,27], which poses a major challenge to the development of a unified cotton LAI canopy spectral index monitoring model in the future. Therefore, this study utilized cotton in the arid area of northwest China as the research object. Based on previous studies, five vegetation indices and spectral reflectance transformation forms were selected to analyze and simulate the correlation of LAI, and an LAI monitoring model of cotton during the entire growth period, bud stage, flowering stage, early boll stage, and full boll stage was constructed to obtain the best LAI monitoring model of cotton in each growth period. The aim of this study was to provide a theoretical basis to accurately monitor the LAI in each growth period of cotton.

## 2. Materials and Methods

### 2.1. Experimental Design

The experiment was conducted at the experimental station of the Key Laboratory of Oasis Ecological Agriculture of Xinjiang Production and Construction Corps (Shihezi, China) (44.18° N, 86.03° E) in 2019. The experimental field had gray desert soil of medium fertility, and its organic matter content was 19.3 g/kg$^{-1}$. The soil nitrogen, total phosphorus, alkali hydrolyzed nitrogen, available phosphorus, and available potassium were 1.l7 g/kg$^{-1}$, 2.31 g/kg$^{-1}$, 75 mg/kg$^{-1}$, 92.5 mg/kg$^{-1}$, and 320 mg/kg$^{-1}$, respectively. A machine-picked cotton planting mode was used (film width 2.05 m, one film, three tubes, six rows, row spacing 66 cm + 10 cm), and the plants were spaced 12.5 cm with two protection rows on each side, drip irrigation under the film, and fertilization with water. The seeds were sown in mid-April using a random block design. The cotton variety selected was Xinluzao 45, which is a representative variety in the Xinjiang cotton-growing area. Five nitrogen treatments were established, and each was conducted in triplicate across a total of 15 plots. The nitrogen treatments were as follows: 0 kg ha$^{-1}$ (N0), 120 kg ha$^{-1}$ (N1; severely deficient in fertilizer), 240 kg ha$^{-1}$ (N2; slightly deficient in fertilizer), 360 kg ha$^{-1}$ (N3; moderately fertilized), and 480 kg ha$^{-1}$ (N4; over-fertilized). Treatment N0 was used as the control. Each plot was 4.6 m wide and 10 m long, which produced a plot area of 46 m$^2$, and isolation zones were placed between the plots. Each cell was used as a sampling point.

Pests and diseases were controlled with pesticides, and other field management measures were conducted following the local conventional high-yield cultivation measures.

### 2.2. Data Collection

#### 2.2.1. Measurement of the Canopy Spectrum

We used an ASD Field Spec Pro FRTM spectrometer (Malvern Panalytical, Malvern, UK) with a wavelength range of 350–2500 nm. The resolution of the spectral region was 3 nm for 350–1000 nm and 10 nm for 1000–2500 nm. The spectral sampling interval was 1 nm, and the field of view was 25°. When collecting spectra in the field, personnel should not wear white or particularly bright clothes to avoid reflection that could affect the collection of the spectra. The cotton canopy spectrum should be measured in the bud stage, flowering stage, early boll stage, and full boll stage. The weather forecast should be studied in advance to select clear and cloudless weather for measurement and avoid cloudy and rainy weather. The time period of solar intensity in Xinjiang is 12:00 to 14:00, which is suitable to measure the spectrum. Plants that grew evenly without diseases and pests were selected for sampling. Before the measurement, the instrument and power supply were turned on, preheated for half an hour in advance, and then the whiteboard was used to calibrate the handheld computer that is connected to the data acquisition via Bluetooth. It was calibrated every 10 times to avoid the "drift" of the measurement results as time progresses, thus reducing its accuracy. The sensor probe points downward, and the optical fiber probe is 50 cm away from the cotton canopy. Ten specification curves were collected at each sampling point. The scan time was set to 0.2 s, and data were collected as the average of the three spectral measurements. However, as 1800–2500 nm spectral information is mainly affected by environmental noise, water, and gas, it is difficult to invert the LAI spectral change rule. In this paper, only 350~1800 nm spectral information is analyzed.

The spectrum acquisition device is shown in Figure 1. It is primarily composed of a tripod, mobile tube, support plate, optical fiber, power supply, and handheld computer. First, the optical fiber was placed in the hollow mobile tube, and the optical fiber probe was fixed vertically downward. The mobile tube can move up and down on the tripod. It primarily adjusts the height of the optical fiber probe and the crop canopy. The canopy height in this experiment was 50 cm. The other end of the optical fiber is connected to the power supply, which is fixed on the support plate of the tripod by nuts. The power supply should be turned down and preheated for half an hour in advance before taking measurements. It is then wirelessly connected through the handheld computer. The parameters are set, the data folders established, the whiteboard calibrated, and the spectra are collected each time. In this study, the sampling points differed each time. As the leaves need to be collected later, the canopy structure is different, and secondary measurement cannot be conducted at the same sampling point.

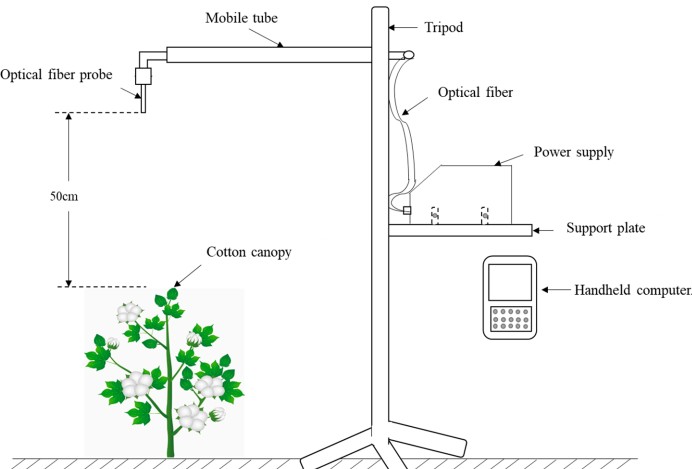

**Figure 1.** The spectrum acquisition device.

2.2.2. Determination of the Leaf Area Index

After the spectral measurement, a ground-based measurement of the cotton leaves was performed using the destruction method. We selected cotton with uniform growth, took two cotton plants at each sampling point, cut out all the leaves, unfolded the leaves, put them into a plastic bag, and recorded the number. Each cotton leaf was separately packed. To prevent the leaf from curling, we put the plastic bag into the fresh-keeping box with ice box. The area of each leaf was measured using an LI-3100 desktop leaf area meter (LICOR, Lincoln, NE, USA). The sampling points differed each time. Because the leaves need to be cut off, the canopy structure is different, and a secondary measurement cannot be conducted at the same sampling point. The LAI was calculated using the following equation:

$$LAI = \frac{K_0 \times N}{K} \tag{1}$$

where LAI is the leaf area index of cotton planted in one measuring plot; $K_0$ is the representative of cotton leaf area in one measurement plot; N is the number of cotton planted in one measuring plot, and K is the area of one measurement plot.

*2.3. Statistical Analysis*

Based on previous studies, this paper selected the normalized difference vegetation index (NDVI) [28], ratio vegetation index (RVI) [29], band-enhanced vegetation index (EVI2) [30], difference vegetation index (DVI), and triangular vegetation index (TVI) to use in this study [31]. These are all vegetation indices that are sensitive to the LAI. Table 1 shows the formulae of the five indices named above. First, the collected spectral data were normalized and then denoised and smoothed. The correlations between the measured cotton LAI and (1) the original canopy spectrum, (2) the first-order differential spectrum, and (3) the logarithm of the reciprocal of the spectrum were determined, and the absolute value of the correlation coefficient was taken. The two largest wavebands were used as the sensitive wavebands for modeling, and the model to estimate the LAI of cotton was established by regression analysis using these sensitive wavelengths.

**Table 1.** The formulae of the five spectral indices that were used to construct the models to estimate the cotton leaf area index.

| Index | Formula |
| --- | --- |
| NDVI | $(R_{nir} - R_{red})/(R_{nir} + R_{red})$ |
| RVI | $R_{nir}/R_{red}$ |
| EVI2 | $2.5 \times (R_{nir} - R_{red})/(R_{nir} + 2.4 \times R_{red} + 1)$ |
| DVI | $R_{nir} - R_{red}$ |
| TVI | $0.5 \times [120 \times (R_{nir} - R_{550}) - 200 \times (R_{red} - R_{550})]$ |

Note: DVI, difference vegetation index; EVI2, band-enhanced vegetation index; NDVI, normalized difference vegetation index; RVI, ratio vegetation index; TVI, triangular vegetation index.

On the one hand, this paper adopted the five vegetation indices that are common to most modern hyperspectral, unmanned aerial vehicles and satellite remote sensing instruments based on previous studies. These included blue, green, red, and NIR. Since the vegetation index established in this paper will be applied to unmanned aerial vehicles and satellite remote sensing in future research, we thought that their use would improve the accuracy of remote sensing monitoring by unmanned aerial vehicles and satellites. Alternatively, these five vegetation indices have unique advantages that are conducive to the analysis of LAI. For example, the NDVI is closely related to photosynthesis and the primary productivity of vegetation. In arid areas, the NDVI can receive a strong influence from the surface environmental factors [32,33]. It can better reflect the changes in LAI of vegetation and is very sensitive to canopy leaf coverage [34]. The RVI is a measure of volume scattering (from randomly oriented indices), a scattering mechanism that is commonly increased by complex structural elements of vegetation, such as a combination

of leaves, branches, and trunks [35]. The EVI2 has substantial advantages in the application of spectral reflectance, which can be used to monitor the vegetation phenology and activities of various ecosystems. It is less sensitive to the environmental background and can improve the monitoring accuracy of LAI [36]. The DVI can effectively reflect the change in vegetation coverage, which is conducive to monitoring the vegetation growth process. It performs well in extracting leaf nutrient content and retrieving vegetation parameters [37]. The TVI can fully reflect the relationship between the radiant energy absorbed by vegetation and the reflectance in red, green, and near-infrared bands by converting the NDVI value of the square root near the Poisson distribution to the normal distribution [38]. The spectral differentiation method is highly sensitive to the spectral signal to noise ratio and can effectively reduce or eliminate the influence of environmental noise, soil background, moisture absorption, and the instrument itself on the accuracy of the model constructed using the original spectra. Differential spectroscopy can obtain accurate vegetation information and monitor the growth status of the vegetation canopy [39]. Ba [40] established a model to estimate the spectral characteristic parameters and canopy fraction of absorbed photosynthetically active radiation (FAPAR) based on the first-order differential spectral reflectance. Sun et al. [41] found that the DVI (435,447) was the optimal spectral index to construct a monitoring model for the LAI of winter wheat. One of the most effective hyperspectral processing technologies in practical applications is first-order differential spectroscopy, which enables changes in the discreteness of hyperspectral data that is measured. The first-order differential spectral transformation generally utilizes the differential method for calculation. The reciprocal logarithm of the original spectrum was utilized to scale the differences between the spectral data in different degrees, so that the different areas were clearer. This eliminates the interference of background noise, degrades the mixed overlapping peaks, and easily locates the band with a high correlation [42–44].

### 2.4. Establishment of the Models

In this paper, we first analyzed the change rule of the LAI and canopy spectrum in different growth periods of cotton under different applications of nitrogen and normalized, denoised, and smoothed the canopy spectrum data. The original spectrum was transformed into first-order differential spectra and the logarithm of the reciprocal spectra. The correlation between the LAI and the original spectrum was analyzed along with the first-order differential spectrum and the logarithm of the reciprocal spectrum. The two bands with the closest correlation were selected as the sensitive bands to establish the NDVI, RVI, EVI2, DVI, and TVI and construct models to monitor the LAI over the different growth stages of cotton and whole growth stages through models of regression analyses. This enabled us to explore the optimal vegetation index monitoring model of the cotton LAI over the different growth stages and whole growth stages to provide a theoretical basis to accurately monitor the changes in the cotton LAI in various growth stages and suggest scientific fertilization.

This study used a total of 280 samples. The first 180 were used for modeling, while the last 100 were used to verify the model. The model testing samples were independent of the modeling data, and 70 samples were collected for each reproductive period, including 45 for modeling and 25 to validate the model. The root mean square error (RMSE), relative error (RE), and correlation coefficient (R) between the predicted and measured values were used to test the accuracy and precision of the model used to estimate the LAI.

The RMSE is the square root of the ratio of the square of the deviation between the predicted value and the true value to the number of observations ($n$). In actual measurement, the number of observations is always limited, and therefore, the true value must be replaced by the most reliable (optimal) value. The standard error is very sensitive to large or small errors in a group of measurements and can adequately reflect the precision of the measurement. Therefore, the standard deviation is used to measure the degree of dispersion

of a set of numbers, and the RMSE is used to measure the deviation between the observed value and the true value. The equation to calculate this is as follows:

$$\text{RMSE} = \sqrt{\frac{\sum_{i=1}^{n}(x_i - y_i)}{n}} \tag{2}$$

where $x_i$ is the actual value; $y_i$ is the model simulation value, and $n$ is the sample size.

The RE refers to the ratio of the absolute measurement error to the measured value expressed as a percentage. In general, the RE can more effectively reflect the credibility of the measurement. For example, measurement result $y$ is subtracted from the agreed true value $t$ of the measured value, and the resulting error or absolute error is Δ. The relative error can then be obtained by dividing the absolute error by the agreed true value. The Pearson's correlation coefficient was the first statistical indicator designed by statistician Carl Pearson. It is a statistical indicator that reflects the degree of linear correlation between variables. The correlation coefficient is calculated according to the product difference method, which is based on the deviation of the two variables from their respective averages. The degree of correlation between the two variables is obtained by multiplying the two deviations. The equation to calculate this is as follows:

$$\text{RE} = \sqrt{\frac{\sum_{i=1}^{n}(x_i - y_i)^2}{\sum_{i=1}^{n} x_i^2}} \tag{3}$$

where $x_i$ is the actual value; $y_i$ is the model simulation value, and $n$ is the sample size.

## 3. Results

### 3.1. Statistical Analysis of LAI of Cotton in Different Growth Stages

The LAI of cotton varies during different growth periods (Table 2), and it also varies under different nitrogen treatments during the same growth period. The LAI reached is maximum value (N3) at the late flowering stage, 4.82. It was at its lowest at the beginning of the Bud stage (N0), 0.39. The lowest standard deviation appeared in the Bolling stage (N1), 0.023, while the highest standard deviation appeared in the flowering stage (N4), 0.936. The lowest coefficient of variation appeared in the Bolling stage (N2), 0.006, and the highest coefficient of variation appeared in the Bud stage (N1), 0.64.

**Table 2.** Statistical analysis of cotton LAI.

| Growth Period | Different Nitrogen Fertilization Treatments | Mean | Min | Max | Standard Deviations | Coefficient of Variation |
|---|---|---|---|---|---|---|
| Bud stage | N0 | 0.87 | 0.39 | 1.61 | 0.531 | 0.612 |
| | N1 | 0.93 | 0.45 | 1.75 | 0.58 | 0.621 |
| | N2 | 0.96 | 0.45 | 1.72 | 0.55 | 0.571 |
| | N3 | 1.02 | 0.48 | 1.72 | 0.515 | 0.505 |
| | N4 | 0.98 | 0.46 | 1.88 | 0.625 | 0.64 |
| Flowering | N0 | 3.31 | 2.15 | 4.05 | 0.823 | 0.249 |
| | N1 | 3.53 | 2.61 | 4.26 | 0.687 | 0.195 |
| | N2 | 3.59 | 2.71 | 4.34 | 0.67 | 0.187 |
| | N3 | 3.78 | 2.59 | 4.82 | 0.919 | 0.243 |
| | N4 | 3.58 | 2.38 | 4.63 | 0.936 | 0.262 |
| Early boll | N0 | 2.86 | 2.68 | 3.06 | 0.169 | 0.059 |
| | N1 | 3.4 | 2.53 | 3.78 | 0.435 | 0.128 |
| | N2 | 2.99 | 2.58 | 3.41 | 0.395 | 0.132 |
| | N3 | 3.33 | 3.11 | 3.55 | 0.203 | 0.061 |
| | N4 | 3.14 | 2.82 | 3.47 | 0.312 | 0.099 |
| Bolling stage | N0 | 2.13 | 2.07 | 2.2 | 0.059 | 0.028 |
| | N1 | 2.34 | 2.31 | 2.37 | 0.023 | 0.01 |
| | N2 | 2.42 | 2.4 | 2.44 | 0.016 | 0.006 |
| | N3 | 2.72 | 2.69 | 2.74 | 0.019 | 0.007 |
| | N4 | 2.39 | 2.34 | 2.44 | 0.044 | 0.018 |

### 3.2. Trend of Variation of the Leaf Area Index (LAI) of Cotton in Different Growth Periods under Different Nitrogen Treatments

The dynamic curve of the LAI showed that the overall change of the LAI under different nitrogen treatments was a low-level single peak change (Figure 2), and different rates of application of nitrogen can control the changes in the LAI throughout the growth period of cotton. The LAI slowly increased at the beginning of the Bud stage under each nitrogen treatment. The increase in treatment N3 was significantly higher than those of the other nitrogen treatments. There were no significant differences between any of the other nitrogen treatments. The LAI rapidly increased from the Bud stage to flowering. At this stage, the size of the LAI followed the order N3 > N2 > N4 > N1 > N0 for the different nitrogen treatments. After July 27, the LAI decreased overall, with the most rapid decrease occurring under treatments N3 and N4, and less rapidly under treatments N0, N1, and N2. The LAI was generally lower for treatments N0, N1, and N4 throughout the growth period. The LAI increases as the rate of application of nitrogen increases. However, nitrogen deficiency or excess will affect the growth of cotton, which results in a decrease in the LAI.

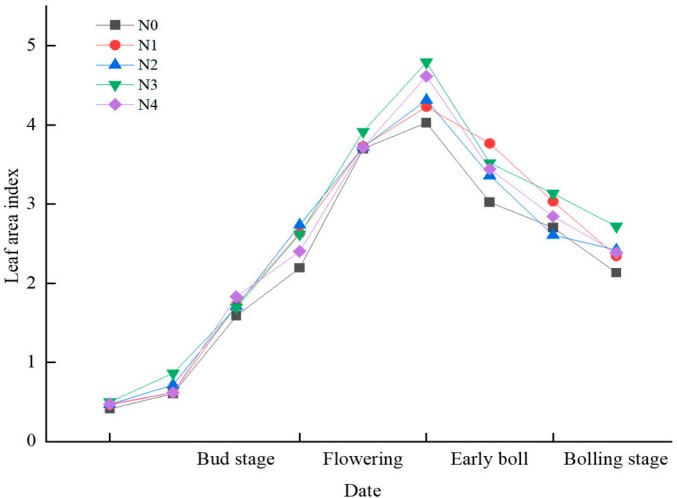

**Figure 2.** Changes in the leaf area index (LAI) of cotton measured under different nitrogen fertilization treatments. NO: 0 kg ha$^{-1}$; N1: 120 kg ha$^{-1}$ (severely deficient in fertilizer); N2: 240 kg ha$^{-1}$ (slightly deficient in fertilizer); N3: 360 kg ha$^{-1}$ (moderate fertilization); N4: 480 kg ha$^{-1}$ (over-fertilization).

The cotton bud stage is the transitional period of growth and development because the growth of the aboveground and belowground parts reaches a certain basis with the increase in temperature, and the plant accelerates and transitions from vegetative to reproductive growth. The reproductive growth begins to be stronger than the vegetative growth. A total of 60 to 80% of the organic matter is transported to the buds, flowers, and bolls for reproductive growth. The LAI reaches its maximum value, which is also the period when cotton requires the most water and fertilizer. During the late flowering and early boll stages of cotton, the competition between vegetative growth and reproductive growth intensifies, leading to the shedding of leaves and a decrease in the LAI.

### 3.3. Changes in the Canopy Spectrum of Cotton at Various Growth Stages under Different Nitrogen Treatments

The changes in the canopy spectrum of cotton at various growth stages under different nitrogen treatments are shown in Figure 3. The changes in the spectral reflectance of cotton under different nitrogen treatments were very similar, and there was a low overall trend.

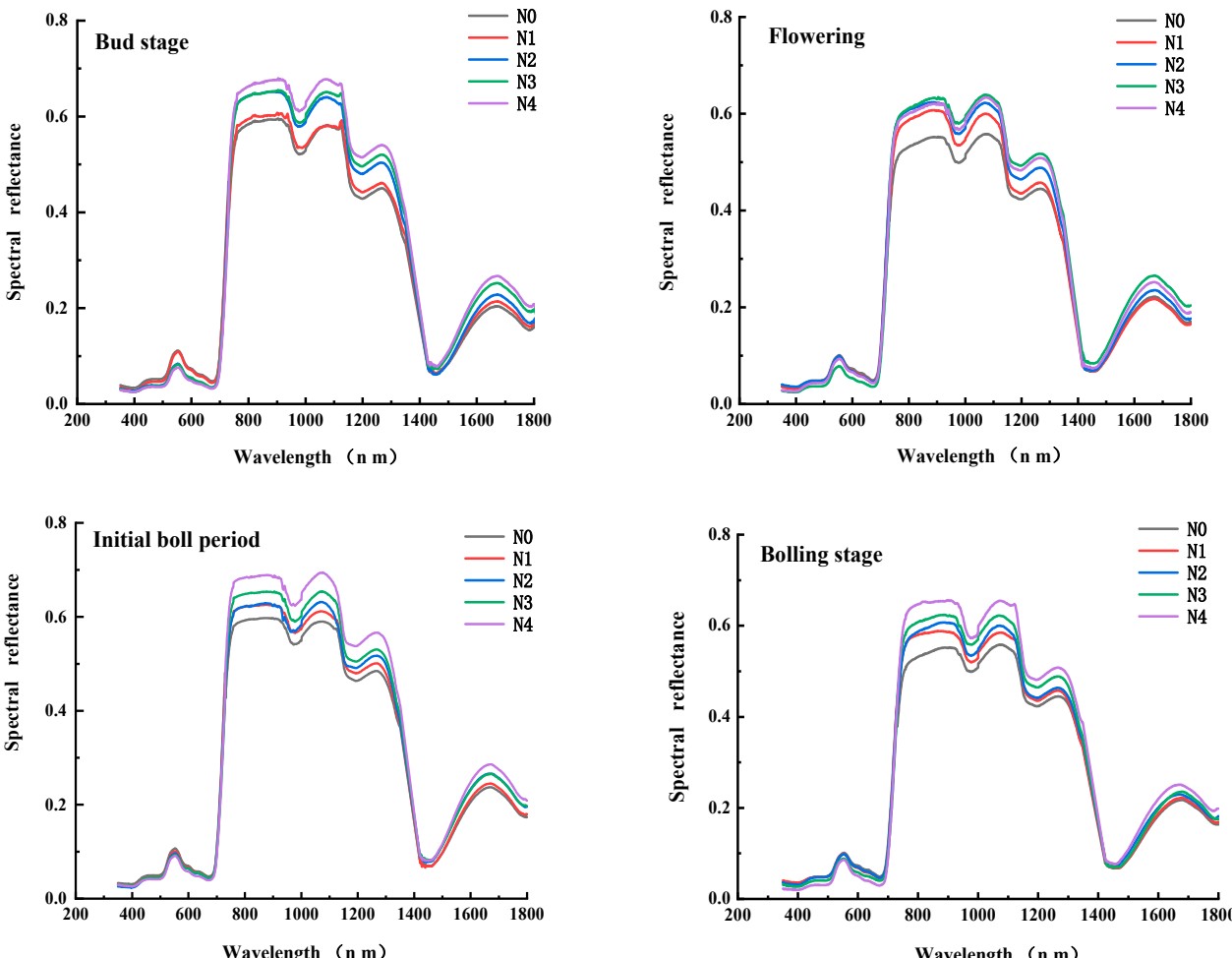

**Figure 3.** The spectral reflectance of the cotton canopy at different growth stages under varying nitrogen treatments.

In the visible light band, the canopy spectral reflectance of cotton at different growth stages was highly similar. The canopy spectral reflectance decreased as the applications of nitrogen increased and followed the order N0 > N1 > N2 > N3 > N4. In the near-infrared band, the canopy spectral reflectance varied significantly in different periods. When the near-infrared band was used in the bud stage, the canopy spectral reflectance can be categorized into a group with relatively low reflectance, which consisted of treatments N0 and N1, and a group with relatively high reflectance, which consisted of treatments N2, N3, and N4. The greatest canopy spectral reflectance was obtained for the highest nitrogen fertilizer content (N4). Moreover, the canopy spectral reflectance followed the order N3 > N4 > N2 > N1 > N0 in the near-infrared band during the flowering stage. The near-infrared band during the early boll and the full boll period indicated that the changes in canopy spectral reflectance under the nitrogen treatments differed compared with the other stages. The canopy spectral reflectance increased with the increase in application of nitrogen, i.e., it followed the order N4 > N3 > N2 > N1 > N0.

### 3.4. Correlation Analysis between the Cotton Canopy Spectral Reflectance and LAI

3.4.1. Correlation Analysis between the Original Canopy Spectrum and LAI

The results of the correlation analysis between the original canopy spectrum and the LAI are shown in Figure 4. The cotton LAI and original canopy reflectance negatively correlated in the visible light band. The smallest correlation coefficient was obtained at 536 nm with an absolute value of 0.5122, while the largest was obtained at 675 nm with

an absolute value of 0.6911. Moreover, there was a positive correlation between the cotton LAI and the original canopy reflectance at 730–1345 nm and a negative correlation at 1346–1800 nm in the near-infrared band. There were two peaks that highly correlated in the whole band. The first peak was at 917 nm, and its correlation coefficient was 0.6452. The second peak appeared at 1067 nm with the largest correlation coefficient of 0.6478. Moreover, there were two valleys at 970 nm and 1200 nm with a correlation coefficient of 0.5332 and 0.2955, respectively. The correlation coefficient was the smallest at 1350 nm, and its absolute value was 0.0119.

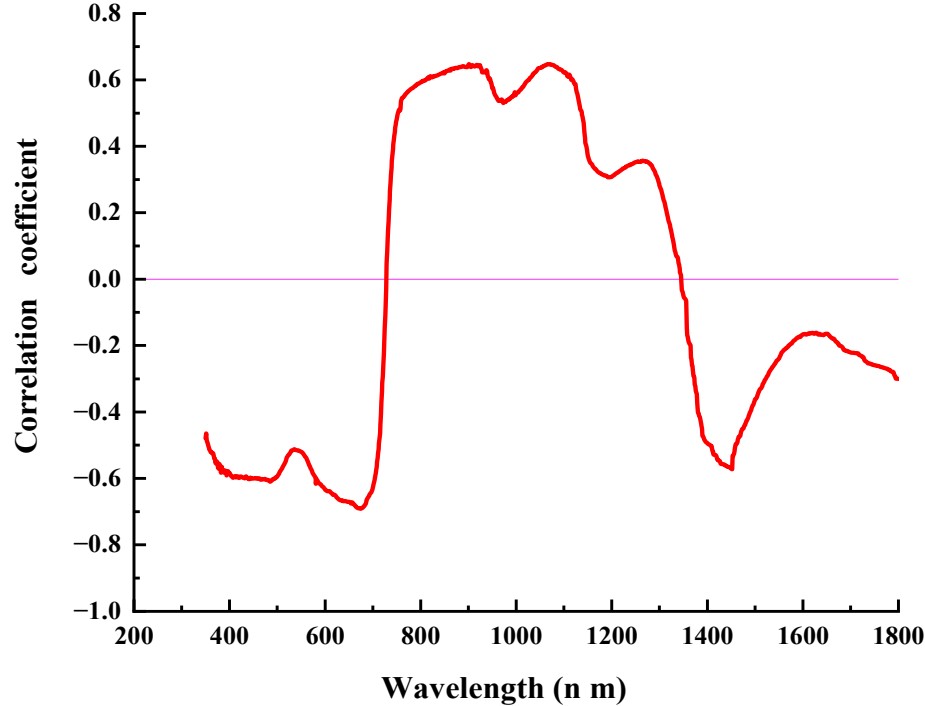

**Figure 4.** The correlation between the canopy spectrum and the leaf area index.

3.4.2. Correlation Analysis between the Original Canopy Spectrum and LAI

The results of the correlation analysis between the canopy first-order differential spectrum and the LAI are shown in Figure 5. The canopy first-order differential spectrum positively correlated with the LAI at approximately 470 nm, and the correlation was relatively high (0.7868). The correlation coefficient at 487 nm was 0.8325, and there was a significant negative correlation at approximately 510 nm. This correlation coefficient was lower than that at 487 nm. At 796 nm, the canopy first-order differential spectrum positively correlated with the LAI with a relatively high value of 0.8598. The first-order differential spectrum of the canopy strongly positively correlated with the LAI at 490 nm, 780 nm, 1030 nm, 1200 nm, and 1425 nm and strongly negatively correlated at 620 nm, 1150 nm, 950 nm, and 1340 nm, indicating that the first-order differential spectrum of the canopy is between the LAI. There are more sensitive bands, and the first-order differential spectrum of the canopy can significantly enhance the spectral information of cotton LAI. The most sensitive band between the first-order differential spectrum of canopy and LAI was at 1142 nm, with the highest correlation of 0.9236. Differential spectroscopy can eliminate the influence of soil background, atmospheric scattering, and absorption on vegetation spectra, which enhances the correlation between spectral reflectance and cotton LAI, and thus improves the accuracy of the model.

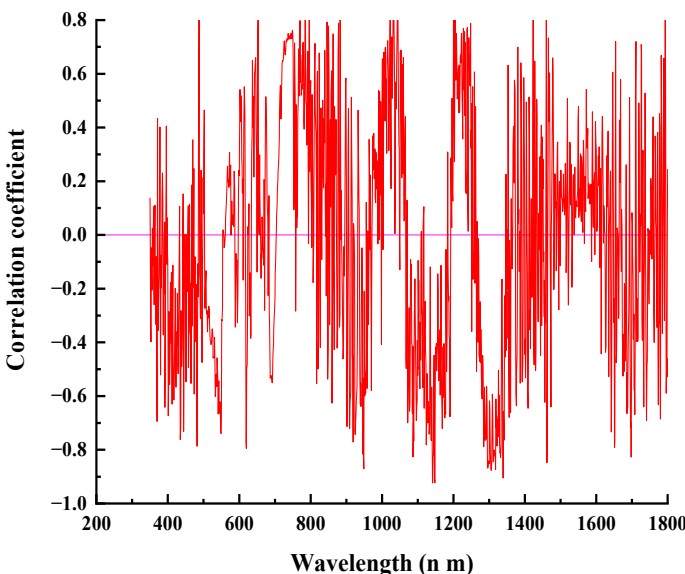

**Figure 5.** The correlation between the canopy first-order differential spectrum and the leaf area index.

### 3.4.3. Correlation Analysis between the Original Canopy Spectrum and LAI

The results of the correlation analysis between the logarithm of the reciprocal of the canopy spectrum and the LAI are shown in Figure 6. The LAI of cotton positively correlated with the logarithm of the reciprocal of the spectrum in the visible light band. There was a trough at 490–580 nm with a minimum correlation coefficient at 536 nm and a value of 0.4761. The highest correlation of any wavelength occurred at 675 nm with a correlation coefficient of 0.6661. There was a negative correlation at 730–1345 nm with a small local peak at 966 nm with a correlation coefficient of −0.5346. The minimum correlation coefficient occurred at 1072 nm with a value of −0.6567. Moreover, there was a positive correlation at 1346–1800 nm, and the largest correlation coefficient in this interval occurred at a peak at 1476 nm with a value of 0.5112. A second peak appeared at 1480 nm with a correlation coefficient of 0.510.

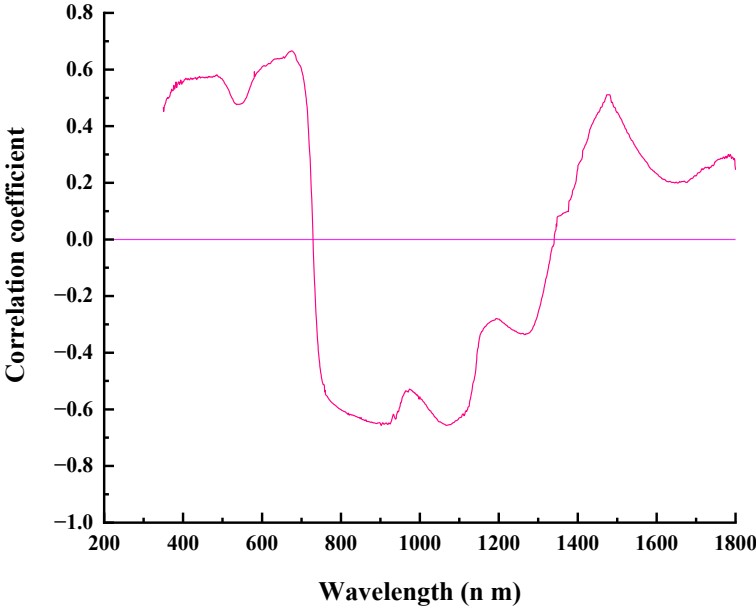

**Figure 6.** The correlation between the logarithm of the reciprocal of the canopy spectrum and the leaf area index.

### 3.5. Construction and Validation of the Canopy Spectral Parameters and the Leaf Area Index Model

3.5.1. Model Establishment

Table 3 shows the details of the models used to estimate the cotton LAI that were constructed using the five spectral indices. The highest $R^2$ values for the original spectrum were obtained for the models built using the RVI and EVI2 with values of 0.86 and 0.84, respectively. The $R^2$ values of the other three models were relatively small. For the first-order differential spectrum, the highest $R^2$ was obtained for the RVI model with a value of 0.8165. Finally, for the logarithm of the reciprocal of the original spectrum, the highest $R^2$ was also obtained for the RVI model with a value of 0.8291. The $R^2$ values of RVI models were relatively high overall, indicating that these models are highly accurate.

**Table 3.** Details of the models used to estimate the leaf area index of cotton during the whole growth period.

| Parameter Name | Parameter Description | Estimation Model | Coefficient of Determination |
|---|---|---|---|
| Original spectrum | NDVI (1067,675)   $(R_{1067} - R_{675})/(R_{1067} + R_{675})$ | $y = 30.47x^2 - 26.70x + 3.542$ | 0.8152 |
| | RVI (1067,675)   $R_{1067}/R_{675}$ | $y = -0.025x^2 + 0.966x - 4.782$ | 0.8611 |
| | EVI2 (1067,675)   $2.5 \times (R_{1067} - R_{675})/(R_{1067} + 2.4 \times R_{675} + 1)$ | $y = -68.19x^2 + 116.8x - 45.73$ | 0.8441 |
| | DVI (1067,675)   $R_{1067} - R_{675}$ | $y = -120.4x^2 + 140.9x - 36.98$ | 0.8354 |
| | TVI (1067,675)   $0.5 \times [120 \times (R_{1067} - R_{550}) - 200 \times (R_{675} - R_{550})]$ | $y = -0.009x^2 - 1.087x - 27.48$ | 0.5272 |
| First-order differential spectrum | NDVI (1142,796)   $(R_{1142} - R_{796})/(R_{1142} + R_{796})$ | $y = -8.224x^2 + 1.362x + 4.182$ | 0.7113 |
| | RVI (1142,796)   $R_{1142}/R_{796}$ | $y = 0.001x^2 - 0.116x + 4.245$ | 0.8165 |
| | EVI2 (1142,796)   $2.5 \times (R_{1142} - R_{796})/(R_{1142} + 2.4 \times R_{796} + 1)$ | $y = 12150x^2 + 742.3x + 5.938$ | 0.6712 |
| | DVI (1142,796)   $R_{1142} - R_{796}$ | $y = -1E+06x^2 - 6291.x - 2.678$ | 0.5882 |
| | TVI (1142,796)   $0.5 \times [120 \times (R_{1142} - R_{550}) - 200 \times (R_{796} - R_{550})]$ | $y = -67.10x^2 + 43.47x - 2.764$ | 0.7695 |
| Logarithm of the reciprocal of the spectrum | NDVI (1072,675)   $(R_{1072} - R_{675})/(R_{1067} + R_{675})$ | $y = -107.8x^2 - 163.7x - 58.00$ | 0.7376 |
| | RVI (1072,675)   $R_{1072}/R_{675}$ | $y = -247.9x^2 + 73.58x - 1.124$ | 0.8291 |
| | EVI2 (1072,675)   $2.5 \times (R_{1072} - R_{675})/(R_{1067} + 2.4 \times R_{675} + 1)$ | $y = -140.9x^2 - 213.7x - 76.93$ | 0.7826 |
| | DVI (1072,675)   $R_{1072} - R_{675}$ | $y = -1.700x^2 - 11.41x - 15.05$ | 0.6165 |
| | TVI (1072,675)   $0.5 \times [120 \times (R_{1072} - R_{550}) - 200 \times (R_{675} - R_{550})]$ | $y = -0.001x^2 + 0.270x - 14.32$ | 0.6133 |

Note: DVI, difference vegetation index; EVI2, band-enhanced vegetation index; NDVI, normalized difference vegetation index; RVI, ratio vegetation index; TVI, triangular vegetation index.

Table 4 shows the construction of the LAI estimation models in the different growth stages of cotton. The sensitive bands of LAI in the different growth stages varied. The sensitive bands in the bud stage were 634 nm, 779 nm, 781 nm, 1363 nm, and 1365 nm. The TVI model constructed by the original spectrum was the highest with an $R^2$ of 0.8979. The flowering sensitive bands were 648 nm, 699 nm, 716 nm, 1239 nm, 1396 nm, and 1402 nm. The NDVI model based on the logarithm of the reciprocal of the spectral index was the highest with an $R^2$ of 0.8746. The sensitive bands at the initial ringing stage were 648 nm, 679 nm, 1239 nm, and 1427 nm. The TVI model based on the logarithm of the reciprocal of the spectral index was the highest with an $R^2$ of 0.8503. The sensitive bands in full boll period were 688 nm, 697 nm, 777 nm, 1133 nm, 1377 nm, and 1399 nm. The DVI model constructed by the original spectrum was the highest with an $R^2$ of 0.8841.

**Table 4.** Construction of the model to estimate the LAI in different growth stages of cotton.

| Parameter Name | Growing Period | Vegetation Index | Estimation Model | Coefficient of Determination |
|---|---|---|---|---|
| Original spectrum | Bud stage | TVI (1365, 779) | $y = -0.0005x^2 - 0.0503x + 0.6039$ | 0.8979 |
| | Flowering | NDVI (1396, 716) | $y = 6.5415x^2 + 6.9377x + 5.0152$ | 0.8161 |
| | Initial boll period | TVI (1427, 679) | $y = -0.0002x^2 + 0.511x + 1.2569$ | 0.7798 |
| | Bolling stage | DVI (1399, 697) | $y = 0.013x^2 - 0.1985x + 0.6913$ | 0.8841 |
| First-order differential spectrum | Bud stage | TVI (1363, 634) | $y = -0.0034x^2 + 0.0768x + 1.4632$ | 0.8816 |
| | Flowering | RVI (1239, 648) | $y = -0.0655x^2 + 0.3474x - 1.4449$ | 0.8017 |
| | Initial boll period | RVI (1239, 648) | $y = -0.0655x^2 + 0.3474x - 1.4449$ | 0.7017 |
| | Bolling stage | TVI (1133, 688) | $y = 47.595x^2 + 16.774x + 4.599$ | 0.7363 |
| Logarithm of the reciprocal of the spectrum | Bud stage | EVI2 (1365, 781) | $y = -1.5303x^2 + 1.6553x + 1.3882$ | 0.8815 |
| | Flowering | NDVI (1402, 699) | $y = 0.0112x^2 - 0.1532x + 0.3088$ | 0.8746 |
| | Initial boll period | TVI (1427, 679) | $y = -0.0014x^2 + 0.1157x + 8.2838$ | 0.8503 |
| | Bolling stage | NDVI (1377, 777) | $y = 0.5334x^2 - 1.4809x + 3.6649$ | 0.8371 |

Note: DVI, difference vegetation index; EVI2, band-enhanced vegetation index; LAI, leaf area index; NDVI, normalized difference vegetation index; RVI, ratio vegetation index; TVI, triangular vegetation index.

### 3.5.2. Model Verification

Table 5 shows the results of verification of the model. As shown in this table, the correlation coefficients between the measured LAI and those estimated by the models were relatively high for the NDVI and RVI models in the original spectrum, with values of 0.8211, and 0.8483, respectively. The RE that was obtained for the NDVI model was relatively small, namely 13.2%. Additionally, the smallest RMSE was also obtained for this model, namely 0.0824. The smallest values of RE and RMSE were both obtained for the RVI model, namely 27.4% and 0.2897, respectively. For the first-order differential spectrum, high correlation coefficients were obtained for the RVI, EVI2, and DVI models, with values of 0.8545, 0.8359, and 0.8772, respectively. The smallest values of RE and RMSE were both obtained for the DVI model, namely 12.4% and 0.1864, respectively. The RE and RMSE of the RVI model were 14.2% and 0.1942, respectively. The RE and RMSE of the EVI2 model were 14.8% and 0.2212, respectively. The highest correlation coefficients for the logarithm of the reciprocal of the original spectrum were obtained for the NDVI, RVI, and DVI models with values of 0.8431, 0.8657, and 0.8366, respectively. The smallest values of RE and RMSE were both obtained for the RVI model, namely 9.29% and 0.1214, respectively. The RE and RMSE of the NDVI model were 10.7% and 0.1556, respectively. The RE and RMSE of the DVI model were 11.9% and 0.1865, respectively. Thus, the RVI model had a good fit between the predicted and measured values, and verification of the accuracy of model was higher than those of the NDVI, DVI, and TVI models.

**Table 5.** The results of the verification of the models to monitor the cotton LAI.

| Parameter Name | | RMSE | RE | Correlation Coefficient |
|---|---|---|---|---|
| Original spectrum | NDVI (1067, 675) | 0.0824 | 13.2 | 0.8211 |
| | RVI (1067, 675) | 0.0797 | 8.9 | 0.8483 |
| | EVI2 (1067, 675) | 0.2533 | 15.7 | 0.7352 |
| | DVI (1067, 675) | 0.2461 | 14.6 | 0.7742 |
| | TVI (1067, 675) | 0.1774 | 13.6 | 0.8052 |
| First-order differential spectrum | NDVI (1142, 796) | 0.2532 | 18.9 | 0.7932 |
| | RVI (1142, 796) | 0.1942 | 14.2 | 0.8545 |
| | EVI2 (1142, 796) | 0.2212 | 14.8 | 0.8359 |
| | DVI (1142, 796) | 0.1864 | 12.4 | 0.8772 |
| | TVI (1142, 796) | 0.2824 | 19.9 | 0.7874 |
| Logarithm of the reciprocal of the spectrum | NDVI (1072, 675) | 0.1556 | 10.7 | 0.8431 |
| | RVI (1072, 675) | 0.1214 | 9.29 | 0.8657 |
| | EVI2 (1072, 675) | 0.2271 | 13.5 | 0.7574 |
| | DVI (1072, 675) | 0.1865 | 11.9 | 0.8366 |
| | TVI (1072, 675) | 0.3475 | 15.9 | 0.7253 |

Note: DVI, difference vegetation index; EVI2, band-enhanced vegetation index; LAI, leaf area index; NDVI, normalized difference vegetation index; RVI, ratio vegetation index; TVI, triangular vegetation index.

Table 6 shows the results of verification of the LAI monitoring model in different growth stages of cotton. The correlation verified by the original spectral TVI (1365,779) model in the bud stage was the highest, and R, RMSE and RE were 0.8137, 0.3127, and 11.29%, respectively. The logarithm NDVI (1402,699) model with the reciprocal of spectral index verified the highest correlation in flowering, and its R, RMSE, and RE were 0.7991, 0.3214, and 5.89%, respectively. The original spectral TVI (1427,679) model was used to verify the highest correlation at the beginning of bolling, and the R, RMSE and RE were 0.8725, 0.2835, and 5.49%, respectively. In the peak boll period, the original spectral DVI (1399,697) model verified the highest correlation, and the R, RMSE, and RE were 0.8633, 0.2932, and 7.07% respectively.

**Table 6.** Validation of the model to monitor the LAI in different growth stages of cotton.

| Parameter | Growth Stage | Vegetation Index | RMSE | RE | Correlation Coefficient |
|---|---|---|---|---|---|
| Original spectrum | Bud stage | TVI (1365, 779) | 0.3127 | 11.29 | 0.8137 |
| | Flowering | NDVI (1396, 716) | 0.4736 | 16.72 | 0.7021 |
| | Initial boll period | TVI (1427, 679) | 0.2835 | 5.49 | 0.8425 |
| | Bolling stage | DVI (1399, 697) | 0.2932 | 7.07 | 0.8633 |
| First-order differential spectrum | Bud stage | TVI (1363, 634) | 0.3584 | 19.55 | 0.7277 |
| | Flowering | RVI (1239, 648) | 0.3925 | 11.25 | 0.7378 |
| | Initial boll period | RVI (1239, 648) | 0.3455 | 10.64 | 0.7917 |
| | Bolling stage | TVI (1133, 688) | 0.3466 | 11.14 | 0.7857 |
| Logarithm of the reciprocal of the spectrum | Bud stage | EVI2 (1365, 781) | 0.3132 | 12.65 | 0.7869 |
| | Flowering | NDVI (1402, 699) | 0.3214 | 5.89 | 0.7991 |
| | Initial boll period | TVI (1427, 679) | 0.3235 | 8.10 | 0.8069 |
| | Bolling stage | NDVI (1377, 777) | 0.3202 | 9.19 | 0.8277 |

Note: DVI, difference vegetation index; EVI2, band-enhanced vegetation index; LAI, leaf area index; NDVI, normalized difference vegetation index; RVI, ratio vegetation index; TVI, triangular vegetation index.

## 4. Discussion

### 4.1. Trends in the Change of Cotton LAI and Canopy Spectra in Different Periods under Varying Nitrogen Treatments

The absorption, transport, and assimilation of nitrogen by cotton leaves in different periods showed dynamic changes. Different rates of N application have obvious regulatory effects on the LAI of cotton during the whole growth period. Compared with other periods, cotton in the bud stage has less demand for nitrogen; the plants and LAI are small, and the plants photosynthesize weakly. There was little difference among the nitrogen treatments. At the flowering stage, the vegetative growth gradually shifted to reproductive growth, and the LAI and photosynthetic capacity increased [45]. The LAI also increased in parallel with the nitrogen, but the N4 treatment started to decrease. Due to the vigorous growth of the plant, the leaves at the lower part of the plant were sheltered from each other, and there was poor ventilation [46]. Thus, the leaves began to fall. At the boll stage, the LAI started to decrease as the growth period advanced. More nutrients were transferred from the leaves to the cotton bolls, which resulted in poor growth and little development of the leaves. They were small or yellow or even fell off. Therefore, the proper application of nitrogen can improve the photosynthetic capacity of leaves, prolong the time for efficient use of light energy, and make cotton reach the most appropriate growth state. The variation coefficient of cotton in the full boll stage was small overall, indicating that there was a stable variation coefficient of cotton LAI; the variation coefficient of bud stage was the largest, and there was a large difference in the cotton LAI. First, cotton growth needs to adapt to the local environment for a period of time during the early stage. Therefore, there is a large difference in plant growth under the influence of environmental factors, such as soil temperature and water. In the late stage, cotton has fully adapted to the local environment and can obtain nutrients to meet its own growth, and the difference in plant growth is reduced.

From the bud stage to the flowering period, the LAI continuously increased from the bud stage to the boll stage). The LAI reached its maximum in the full boll stage. As the growth period advanced, and the content of nitrogen increased, the LAI gradually increased at a slow and non-significant rate during the early stage, and the difference was more apparent in the later stage. This is consistent with the results of Wen [47]. After entering the boll stage, the leaves began to wilt and fall, which resulted in an overall decrease in the LAI. The LAI of various nitrogen treatments during the entire growth stage followed the order N2 > N3 > N4 > N1 > N0. The LAI at N0 was low during the entire period, with a small range of variation. The N3 and N4 treatments most quickly decreased the LAI during the boll stage. Excessive fertilization leads to the excessive growth of cotton. The mutual shielding of the leaves, poor ventilation, and an early plant undergoes senescence and sheds its leaves, resulting in a rapid decrease in the LAI.

The content of the biochemical components of the leaves also changed continuously, which resulted in changes in leaf color, shape, size, and morphological structure. Thus, they affected the absorption, reflection, and transmission of the spectrum. These changes were reflected in the spectral characteristics. In the visible light range with an increase in the rate of nitrogen applied, the spectral reflectance of the canopy decreased following the order N0 > N1 > N2 > N3 > N4, which is consistent with the results of Read et al. [48] and Wu et al. [49]. In the visible light range, the LAI and chlorophyll content increased with the growth stages of cotton, which resulted in a lower canopy spectral reflectance of N4 and a higher canopy spectral reflectivity of N0. The spectral reflectance of the cotton canopy in the near infrared band changed greatly in all the nitrogen treatments, which was consistent with the results of Wu et al. [50]. Cotton has less demand for nitrogen in the bud stage. In this stage, the canopy spectral reflectance in the near-infrared band could be categorized into a group with relatively low reflectance. It consisted of treatments N0 and N1 and a group with relatively high reflectance, which consisted of treatments N2, N3, and N4. During the flowering period, the cotton increased its demand for nitrogen. In the near-infrared band, the canopy spectral reflectance generally increased with an increase in the application of nitrogen. However, the canopy spectral reflectance reached its maximum when the appropriate rate of nitrogen was applied (N3). In contrast, the canopy spectral reflectance decreased when excessive amounts of nitrogen were applied (N4). During this period, the cotton was in the fertile stage of reproductive and vegetative growth, and the optimal rate of nitrogen applied had the most apparent effect on canopy structure. The canopy spectral reflectance differed under different nitrogen treatments during the early and peak bolling stages in the near-infrared band. The canopy spectral reflectance increased with increasing nitrogen application, following the order N4 > N3 > N2 > N1 > N0. This is consistent with the results of Wang et al. [51]. The boll stage of cotton is the most vigorous period of vegetative growth, and it is also the most vigorous period of the growth of young buds and young bolls. If the plant lacks fertilizer, the vegetative and reproductive organs will compete for it. This results in a higher transfer of more nutrients to the vegetative organs. The transfer of nutrients from the leaves to the boll leads to poor growth and development, thin plants, and small and yellow leaves or leaves that fall, thereby reducing the spectral reflectance of the plant.

In this study, the overall spectral reflectance of the cotton canopy was found to be relatively high. This differed from the results of Neale et al. [52], who used spectral data to monitor cotton in northern Xinjiang and found that the spectral reflectance of cotton was relatively low during the entire growth period. This difference may be related to the time at which the spectrum was measured and the cotton variety.

### 4.2. Correlation Analysis between Cotton Canopy Spectral Reflectance and LAI

In this study, in addition to the moisture absorption band spectrum, the correlation analysis showed that the LAI negatively correlated with the original spectral reflectance in the visible light band, which positively correlated at 730–1345 nm and negatively correlated at 1346–1800 nm, which is similar to the results of Bai et al. [53] for cotton. The results

indicate that the original spectral reflectance correlated the most strongly with the LAI at 675 and 1067 nm, with absolute values of the correlation coefficient of 0.6911 and 0.6452, respectively. The first-order differential spectrum positively correlated with the LAI at 796 nm, with a relatively high correlation coefficient of 0.8598 and negatively correlated at 1142 nm with the highest absolute value of the correlation coefficient of 0.9236. Qi et al. [54] found that the band that was the most sensitive to the cotton canopy LAI in the original spectral reflectance, which appeared at 1461 nm, and its correlation coefficient was 0.726. Additionally, the same researchers found that the band that was the most sensitive to the cotton canopy LAI for the first-order differential spectrum appeared at 742 nm with a correlation coefficient of 0.744. It is the difference in photosynthesis and chlorophyll content of crops in different regions that leads to deviations in spectral reflectance. Moreover, the LAI positively correlated with the logarithm of the reciprocal of the spectrum in the visible band. The highest correlation was at 675 nm with a correlation coefficient of 0.6661. There was a negative correlation at 730–1345 nm with the absolute value of the correlation coefficient the highest at 1072 nm and a value of 0.6567. There was a positive correlation at 1346–1800 nm. The correlation coefficients for the original spectrum and the logarithm of the reciprocal of the spectrum are generally opposite, which is consistent with the results of Wang et al. [55].

*4.3. Construction and Verification of the LAI Model Based on Canopy Spectrum and Spectral Indices*

The highest $R^2$ values for the original spectra were obtained for the models that were constructed using the RVI and EVI2. Jin et al. [56] found that the NDVI (890, 670) had the strongest correlation with the cotton LAI of the spectral indices tested. The difference between the findings of Jin et al. [56] and those of this study may be related to differences in the climatic conditions, solar radiation, and solar altitude at the time of measurement, which could have resulted in different spectral sensitivities and model accuracy. In this study, the most accurate model for the first-order differential spectrum and the logarithm of the reciprocal of the spectrum was obtained for the RVI model with $R^2$ values that ranged from 0.8165 to 0.8611, while the second most accurate model was obtained for the NDVI model. This is consistent with the findings of Tůma et al. [57] for wheat. In this study, the RVI model produced the highest $R^2$ values for the original spectrum, the first-order differential spectrum, and the logarithm of the reciprocal of the spectrum with values between 0.8453 and 0.8657. This study shows that the RVI had a higher fitting accuracy during the modeling process and a smaller error in model testing compared with the other four spectral indices that were studied. Therefore, the RVI can further improve the accuracy and precision of the estimation of LAI during the entire cotton growth period and provide a theoretical basis and technical approach for the remote sensing-based monitoring of the crop LAI, which is similar to the results of Li et al. [58].

The monitoring models of LAI varied across the different growth stages of cotton. The bud and early boll stages were consistent with the TVI model. The flowering stage was the NDVI model, and the full boll stage was the DVI model. These differences fully show the necessity of modeling by growth stages. Experiments that only determine the sensitive band of the entire growth stage cannot accurately reflect the characteristics of cotton LAI in different growth stages, and the values of the same vegetation index of the same crop in different growth stages vary. The correlation with agronomic parameters also differed. Therefore, determining the relationship between the vegetation index in different growth periods of cotton and the LAI in corresponding periods is the primary condition for spectral monitoring and diagnosis. In the later stage, the changes of LAI in cotton can be monitored based on the models constructed in different periods to improve the monitoring accuracy of LAI in each growth period of cotton.

This paper monitored the cotton LAI based on ground hyperspectral readings. There was high-ground hyperspectral resolution highly accurate monitoring. Compared with the UAV and satellite remote sensing, the monitoring range was limited [59]. It is difficult to

obtain a wide range of crop growth conditions, and the readings need to be measured in a clear and cloudless condition. If they are collected in cloudy weather or in the morning and afternoon, the transmission, reflection, and absorption of light will be affected, and the collected spectral information will be highly inadequate. The reduction of sensitive spectral information makes it difficult to retrieve the vegetation parameter information, and it is difficult to find crops with insufficient nutrition and pests in time, which leads to the desiccation of plants or even death in the later stage [60]. Therefore, In future research, the optimal vegetation index established by ground hyperspectral in each growth period can be used in UAV and satellite remote sensing. The UAV and satellite remote sensing data can be used to simulate the ground spectrum based on the spectral response function. Through cloud removal, noise removal, spectral transformation and other spectral preprocessing techniques and in-depth learning methods, the monitoring accuracy of UAV and satellite remote sensing can be improved, which also improves the ground hyperspectral and UAV complementary of the spectral information provided by satellite remote sensing. besides, nitrogen has a significant regulatory effect on the LAI of cotton during the whole growth period. Monitoring the LAI can indirectly reflect the crop nutrition status and provide further guidance for experts to conduct and diagnose the levels of nutrition and recommend fertilization programs for different growth periods of cotton.

**5. Conclusions**

In this study, the use of sensitive bands enabled the analysis of correlation between cotton canopy spectral reflectance and LAI in different growth periods. The use of these bands led to the establishment of a cotton LAI monitoring model and tests of the models. The main conclusions are as follows:

1.  Different nitrogen treatments led to differences in the LAI of cotton in each growth stage. The changes in the cotton canopy spectral reflectance and the LAI differed in each period. The canopy spectral reflectance and the LAI showed an overall single-peak change of low and high. In the visible light range, the canopy spectral reflectance decreased with increasing fertilization in all the growth stages. In the near-infrared band, the canopy spectral reflectance increased with an increase in the rate of application of nitrogen in the bud stage, early boll stage, and full boll stage. However, the spectral reflectance was the maximum for the second-highest fertilization amount (N3) in the flowering stage, and the canopy spectral reflectance was low for the severe fertilizer shortage (N0) and excessive application of fertilizer (N4). These results suggest that spectral remote sensing can be used to determine optimal amounts of fertilization and achieve the real-time monitoring of agricultural conditions.

2.  The sensitive bands of LAI varied in different growth stages of cotton. The bands of the original spectral reflectance that were the most sensitive to the cotton LAI were 675 and 1067 nm, and the bands at which the logarithm of the reciprocal of the spectrum were the most sensitive to the cotton LAI were 675 and 1072 nm. The distribution diagrams of the two are opposite. For the first-order differential spectrum, the bands that were the most sensitive to the cotton LAI were 796 and 1142 nm.

3.  The vegetation index monitoring models constructed by cotton LAI in different growth stages differed. The TVI model was the highest during the bud stage and early boll stage, and its $R^2$ values were 0.8137 and 0.8725. The NDVI model was the highest during the flowering stage with an $R^2$ of 0.7991, and the DVI model was the highest in the full boll stage with an $R^2$ of 0.8633. The RVI model constructed by cotton LAI during the entire growth period was the most accurate. The model has minimal error and is sensitive to the change of cotton LAI during the entire growth period. It can serve as one of the best models to monitor the change in cotton LAI.

**Author Contributions:** Conceptualization, X.F. and X.L.; methodology, X.F. and P.G.; software, L.Z.; validation, Z.Z., L.Z. and Q.Z.; formal analysis, Q.Z. and L.M.; investigation, Y.M. and X.Y.; resources, X.L.; data curation, C.Y.; writing—original draft preparation, X.F.; writing—review and editing, X.F.; visualization, C.Y.; supervision, X.L.; project administration, X.L.; funding acquisition, X.L. All authors have read and agreed to the published version of the manuscript.

**Funding:** This research was funded by the Innovative Team Project in the Key Fields of Xinjiang Production and Construction Corps (2018CB004), the International Cooperation Project of Xinjiang Production and Construction Corps (2018BC009), the General Funded Projects of China Postdoctoral Science Foundation (2017M623282), and the Innovation Development Project of Shihezi University (CXFZ201903).

**Conflicts of Interest:** The authors declare no conflict of interest.

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
