# Peer review of "Establishment of a Monitoring Model for the Cotton Leaf Area Index Based on the Canopy Reflectance Spectrum"

_land, doi:10.3390/land12010078_

Round 1

Reviewer 1 Report

The paper entitled "Establishment of a Monitoring Model for the Cotton Leaf Area Index Based on the Canopy Reflectance Spectrum" concerns field measurements of spectral reflectance at cotton test site for providing modelled vegetation index (LAI). My research unit I have been working is doing very similar measurements with ASD FieldSpec over winter wheat sites. Thus I think it is quite interesting for me which vegetation indices the Author(s) assessed decribing reference LAI best.

I have some minor remarks to be included in the manuscript. My remarks and comments concern mainly introduction, methodology and discussion. Here you find the notes listed below as follows:

1) Introduction
Referring to line [37] you wrote  "For example, the LAI has been found to significantly correlate with crop yield". Please add some references that confirm your statement. I suggest the next references concerning remote-sensing and field-based LAI for modelling yields at different crop types across many countries in Europe, Asia and North America.

Ji, Z.; Pan, Y.; Zhu, X.; Wang, J.; Li, Q. Prediction of Crop Yield Using Phenological Information Extracted from Remote Sensing Vegetation Index. Sensors 2021, 21, 1406. https://doi.org/10.3390/s21041406

Zhu, X.; Guo, R.; Liu, T.; Xu, K. Crop Yield Prediction Based on Agrometeorological Indexes and Remote Sensing Data. Remote Sens. 2021, 13, 2016. https://doi.org/10.3390/rs13102016

K. Dabrowska-Zielinska et al., "Crop Yield Modelling Applying Leaf Area Index Estimated from Sentinel-2 and Proba-V Data at JECAM site in Poland," IGARSS 2018 - 2018 IEEE International Geoscience and Remote Sensing Symposium, 2018, pp. 5382-5385, doi: 10.1109/IGARSS.2018.8519120.

Mokhtari, Ali & Noory, Hamideh & Vazifedoust, M.. (2018). Improving crop yield estimation by assimilating LAI and inputting satellite-based surface incoming solar radiation into SWAP model. Agricultural and Forest Meteorology. 250. 159-170. 10.1016/j.agrformet.2017.12.250.

2) Introdution
Referring to line [74] you wrote:
"Based on previous studies, five vegetation indices and spectral reflectance transformation forms were selected to analyze and simulate the correlation of LAI, and an LAI monitoring model of cotton during the entire growth period, bud stage, flowering stage, early boll stage and full boll stage was constructed to obtain the best LAI monitoring model of cotton in each growth period."

I am just thinking why five vegetation indices only you have been taken into account? Please see very interesting database available at the link https://www.indexdatabase.de/db/i.php concerning over 500(!) vegetation indices might be used for monitoring crop (cotton) stages. What are advantages and backwards of the popular and commonly used indices you have chosen? Why didn't you take more of them?

3) Materials and methods
Referring to line [116] on "Determination of the leaf area index".
Please add the table on variation of the reference LAI measurements.  Such results as mean and standard deviations would be sufficient for emphasising the quality of samples done under LI-3000 measurements.

4) Discussion
I think you should add some sentences on aerial/satellite hyperspectral missions that could be applicable for monitoring LAI over cotton at medium scales (EnMAP, HyMap). Aerial and satellite-based images are powerful source of spatio-temporal information on hyperspectral reflectance that might be used for incorporating your model equations from Table no.4 and Table no.5.

5) Discussion
Your results revealed quite strong correlations of field reflectance measurements with referenced LAI. I think you should add more information on weather conditions that might affect the quality of spectral measurements. What about frequent cloud observations? Does your modelled LAI fits properly cotton stages for monitoring under cloudy observations?

Thank you.

Author Response

The paper entitled "Establishment of a Monitoring Model for the Cotton Leaf Area Index Based on the Canopy Reflectance Spectrum" concerns field measurements of spectral reflectance at cotton test site for providing modelled vegetation index (LAI). My research unit I have been working is doing very similar measurements with ASD FieldSpec over winter wheat sites. Thus I think it is quite interesting for me which vegetation indices the Author(s) assessed decribing reference LAI best.

I have some minor remarks to be included in the manuscript. My remarks and comments concern mainly introduction, methodology and discussion. Here you find the notes listed below as follows:

Response: Thank you for your interest in our research. We introduced the necessary improvements based on your suggestions.

1) Introduction

Referring to line [37] you wrote  "For example, the LAI has been found to significantly correlate with crop yield". Please add some references that confirm your statement. I suggest the next references concerning remote-sensing and field-based LAI for modelling yields at different crop types across many countries in Europe, Asia and North America.

Ji, Z.; Pan, Y.; Zhu, X.; Wang, J.; Li, Q. Prediction of Crop Yield Using Phenological Information Extracted from Remote Sensing Vegetation Index. Sensors 2021, 21, 1406. https://doi.org/10.3390/s21041406

Zhu, X.; Guo, R.; Liu, T.; Xu, K. Crop Yield Prediction Based on Agrometeorological Indexes and Remote Sensing Data. Remote Sens. 2021, 13, 2016. https://doi.org/10.3390/rs13102016

  1. Dabrowska-Zielinska et al., "Crop Yield Modelling Applying Leaf Area Index Estimated from Sentinel-2 and Proba-V Data at JECAM site in Poland," IGARSS 2018 - 2018 IEEE International Geoscience and Remote Sensing Symposium, 2018, pp. 5382-5385, doi: 10.1109/IGARSS.2018.8519120.

Mokhtari, Ali & Noory, Hamideh & Vazifedoust, M.. (2018). Improving crop yield estimation by assimilating LAI and inputting satellite-based surface incoming solar radiation into SWAP model. Agricultural and Forest Meteorology. 250. 159-170. 10.1016/j.agrformet.2017.12.250.

Response: Thank you for your advice. We added the references that the reviewer kindly provided (Lines 39 of the revised manuscript).

2) Introdution

Referring to line [74] you wrote:

"Based on previous studies, five vegetation indices and spectral reflectance transformation forms were selected to analyze and simulate the correlation of LAI, and an LAI monitoring model of cotton during the entire growth period, bud stage, flowering stage, early boll stage and full boll stage was constructed to obtain the best LAI monitoring model of cotton in each growth period."

I am just thinking why five vegetation indices only you have been taken into account? Please see very interesting database available at the link https://www.indexdatabase.de/db/i.php concerning over 500(!) vegetation indices might be used for monitoring crop (cotton) stages. What are advantages and backwards of the popular and commonly used indices you have chosen? Why didn't you take more of them?

Response: We are very grateful to the reviewer's concern on this issue. We added the relevant explanation. On the one hand, this paper adopted the five vegetation indices that are common to most modern hyperspectral, unmanned aerial vehicles and satellite remote sensing instruments based on previous studies. These included blue, green, red and NIR. Since the vegetation index established in this paper will be applied to unmanned aerial vehicles and satellite remote sensing in future research, we thought that their use would improve the accuracy of remote sensing monitoring by unmanned aerial vehicles and satellites. Alternatively, these five vegetation indices have unique advantages, they are conducive to the analysis of LAI. For example, the NDVI is closely related to photosynthesis and the primary productivity of vegetation. In arid areas, the NDVI can receive a strong influence from the surface environmental factors. It can better reflect the changes in LAI of vegetation and is very sensitive to canopy leaf coverage. The RVI is a measure of volume scattering (from randomly oriented indices), a scattering mechanism that is commonly increased by complex structural elements of vegetation, such as a combination of leaves, branches, and trunks. The EVI2 has substantial advantages in the application of spectral reflectance, which can be used to monitor the vegetation phenology and activities of various ecosystems. It is less sensitive to the environmental background and can improve the monitoring accuracy of LAI. The DVI can effectively reflect the change in vegetation coverage, which is conducive to monitoring the vegetation growth process. It performs well in extracting leaf nutrient content and retrieving vegetation parameters. The TVI can fully reflect the relationship between the radiant energy absorbed by vegetation and the reflectance in red, green ,and near-infrared bands by converting the NDVI value of the square root near the Poisson distribution to the normal distribution. (Lines 206-228 of the revised manuscript)

3) Materials and methods

Referring to line [116] on "Determination of the leaf area index".

Please add the table on variation of the reference LAI measurements.  Such results as mean and standard deviations would be sufficient for emphasising the quality of samples done under LI-3000 measurements.

Response: Thank you for your advice. We added the table on the changes in LAI change to the manuscript as suggested by the reviewer. The LAI of cotton varies during different growth periods (Table 2), and it also varies under different nitrogen treatments during the same growth period. The LAI reached is maximum value (N3) at the late flowering stage, 4.82. It was at its lowest at the beginning of Bud stage (N0), 0.39. The lowest standard deviation appeared in the Bolling stage (N1), 0.023, while the highest standard deviation appeared in the flowering stage (N4), 0.936. The lowest coefficient of variation appeared in the Bolling stage (N2), 0.006, and the highest coefficient of variation appeared in the Bud stage (N1), 0.64. (Lines 298-306 of the revised manuscript)

Table 2. Statistical analysis of the cotton leaf area index.

Growth period

Different nitrogen fertilization treatments

Mean

Min

Max

Standard deviations

Coefficient of Variation

N0

0.87

0.39

1.61

0.531

0.612

N1

0.93

0.45

1.75

0.58

0.621

Bud stage

N2

0.96

0.45

1.72

0.55

0.571

N3

1.02

0.48

1.72

0.515

0.505

N4

0.98

0.46

1.88

0.625

0.64

N0

3.31

2.15

4.05

0.823

0.249

N1

3.53

2.61

4.26

0.687

0.195

Flowering

N2

3.59

2.71

4.34

0.67

0.187

N3

3.78

2.59

4.82

0.919

0.243

N4

3.58

2.38

4.63

0.936

0.262

N0

2.86

2.68

3.06

0.169

0.059

N1

3.4

2.53

3.78

0.435

0.128

Early boll

N2

2.99

2.58

3.41

0.395

0.132

N3

3.33

3.11

3.55

0.203

0.061

N4

3.14

2.82

3.47

0.312

0.099

N0

2.13

2.07

2.2

0.059

0.028

N1

2.34

2.31

2.37

0.023

0.01

Bolling stage

N2

2.42

2.4

2.44

0.016

0.006

N3

2.72

2.69

2.74

0.019

0.007

N4

2.39

2.34

2.44

0.044

0.018

4) Discussion

I think you should add some sentences on aerial/satellite hyperspectral missions that could be applicable for monitoring LAI over cotton at medium scales (EnMAP, HyMap). Aerial and satellite-based images are powerful source of spatio-temporal information on hyperspectral reflectance that might be used for incorporating your model equations from Table no.4 and Table no.5.

Response: We are very grateful to the reviewer's concern on this issue. We added the relevant explanation. The estimation of accuracy based on ground hyperspectral readings in this study was high, but whether it can be applied in UAV and at the satellite scale still merits further exploration. Although the ground hyperspectral readings have a high resolution and are highly accurate at monitoring, they are limited for monitoring large-scale farmland crops. However, UAV and satellite remote sensing can quickly obtain large-scale farmland crop information in a timely manner, but the model is not very accurate. Therefore, the optimal vegetation index established by the ground hyperspectral readings in each growth period can be used in UAV and satellite remote sensing in the later stage. The UAV and satellite remote sensing data were used to simulate the ground spectrum based on the spectral response function, and the monitoring accuracy of UAV and satellite images was improved through such spectral preprocessing technologies as cloud removal, noise removal, spectral transformation, and in-depth learning methods. This enabled the realization of the complementation of ground hyperspectral and UAV and satellite remote sensing spectral information, and it provides a theoretical basis to obtain a wide range of vegetation growth status and a model to monitor the LAI of cotton during each growth period. (Lines 624-641 of the revised manuscript)

5) Discussion

Your results revealed quite strong correlations of field reflectance measurements with referenced LAI. I think you should add more information on weather conditions that might affect the quality of spectral measurements. What about frequent cloud observations? Does your modelled LAI fits properly cotton stages for monitoring under cloudy observations?

Response: Thank you for your advice. We added the relevant explanation. In this study, the canopy spectra were obtained under sunny and cloudless conditions and direct sunlight at noon. However, when there are clouds or moisture that collects in the morning and afternoon, this will affect the transmission, reflection and absorption of light. The collected spectral information is quite different, and it is difficult to retrieve the vegetation parameter information. In future research, we considered using an optimal vegetation index established by the ground hyperspectral readings in each growth period in the UAV and satellite images. Increased satellite remote sensing of different cloud amounts should be used to monitor and evaluate the vegetation and further explore the monitoring effect of the same vegetation index in different growth periods on different spatial scales. (Lines 626-641 of the revised manuscript)

Reviewer 2 Report

1. Line105-115 How to select cotton population when collecting cotton canopy spectrum? Is the cotton canopy structure growing differently or evenly? Please elaborate. 

2. How to avoid the influence of human factors when collecting spectral information in the field? (What are the specific requirements for the sampler?) 

3. Line117-127 LAI of cotton is measured by destructive method, but will this affect the next sampling? Is each sampling based on the same point? Please elaborate. 

4. After the cotton leaves are collected, their LAI needs to be measured. This article introduces that LAI is measured indoors. Will the leaves change during the period from field to indoor? Does the change affect the determination of LAI? 

5. Please specify the specific number of samples for model construction and verification in this paper 

6. The spectral instrument described in this paper has a band range of 350nm-2500nm, but why is only 350-1800nm shown in Figure 2? 

7. Vegetation index is constructed based on sensitive band. How to select the sensitive band of vegetation index in this paper? Please elaborate.

Author Response

  1. 1. Line105-115 How to select cotton population when collecting cotton canopy spectrum? Is the cotton canopy structure growing differently or evenly? Please elaborate.

Response: We are very grateful to the reviewer's concern on this issue. We added the relevant explanation. The cotton canopy spectrum should be measured in the bud stage, flowering stage, early boll stage and full boll stage. The weather forecast should be studied in advance to select clear and cloudless weather for measurement and avoid cloudy and rainy weather. The time period of solar intensity in Xinjiang is 12:00 to 14:00, which is suitable to measure the spectrum. Plants that grew evenly without diseases and pests were selected for sampling. Before the measurement, the instrument and power supply were turned on, preheated for half an hour in advance, and then the whiteboard was used to calibrate the handheld computer that is connected to the data acquisition through Bluetooth. It was calibrated every 10 times to avoid the "drift" of the measurement results as time progresses, thus, reducing its accuracy. The sensor probe points downward, and the optical fiber probe is 50 cm away from the cotton canopy. Ten specification curves were collected at each sampling point. The scan time was set to 0.2 seconds, and data were collected as the average of the three spectral measurements. (Lines 142-156 of the revised manuscript)

  1. How to avoid the influence of human factors when collecting spectral information in the field? (What are the specific requirements for the sampler?)

Response: When collecting spectra in the field, the personnel should not wear white or particularly bright clothes to avoid reflection that could affect the collection of spectra. This paper utilized a spectral acquisition device (Fig.1). It is primarily composed of a tripod, mobile tube, support plate, optical fiber, power supply, and handheld computer. First, the optical fiber was placed in the hollow mobile tube, and the optical fiber probe was fixed vertically downward. The mobile tube can move up and down on the tripod. It primarily adjusts the height of the optical fiber probe and the crop canopy. The canopy height in this experiment was 50 cm. The other end of the optical fiber is connected to the power supply, which is fixed on the support plate of the tripod by nuts. The power supply should be turned down and preheated for half an hour in advance before taking measurements. It is then wirelessly connected through the handheld computer. The parameters are set; the data folders established, the whiteboard calibrated, and the spectra were collected each time. The sampling points differed each time. Because the leaves need to be collected later, the canopy structure is different, and secondary measurement cannot be conducted at the same sampling point. (Lines 140-142 and 158-172 of the revised manuscript)

Fig. 1.  The spectral acquisition device

  1. Line117-127 LAI of cotton is measured by destructive method, but will this affect the next sampling? Is each sampling based on the same point? Please elaborate.

Response: In order to better understand the blade sampling information, we have added the relevant explanation, the original sentence Additionally, ground-based measurement of the cotton leaves was performed using the destruction method. After the spectral measurements were performed, samples were immediately removed from the measurement area. Three cotton plants were sampled from each sample point. The leaves were then removed, and each leaf was placed in a separate sealed box and immediately sent to the laboratory for analysis. The area of each leaf was measured using an LI-3100 desktop leaf area meter (LICOR, Lincoln, NE, USA). have been rewritten to “ After the spectral measurement, ground based measurement of the cotton leaves was performed using the destruction method. We selected cotton with uniform growth, took two cotton plants at each sampling point, cut out all the leaves, unfolded the leaves, put them into a plastic bag, and recorded the number. Each cotton leaf was separately packed. To prevent the leaf from curling, put the plastic bag into the fresh-keeping box with ice box, The area of each leaf was measured using an LI-3100 desktop leaf area meter (LICOR, Lincoln, NE, USA). The sampling points differed each time. Because the leaves need to be cut off, the canopy structure is different, and secondary measurement cannot be conducted at the same sampling point. (Line 180-188of revised manuscript) .

  1. After the cotton leaves are collected, their LAI needs to be measured. This article introduces that LAI is measured indoors. Will the leaves change during the period from field to indoor? Does the change affect the determination of LAI?

Response: When we collect the leaves, Each cotton leaf was separately packed. To prevent the leaf from curling, put the plastic bag into the fresh-keeping box with ice box, The leaves will not change and will not affect the determination. (Line 182-185of revised manuscript).

  1. Please specify the specific number of samples for model construction and verification in this paper .

Response: Thanks for carefully reviewing the manuscript. We have added the relevant explanation. A total of 280 samples were collected in this experiment. In order to ensure the uniformity of the sample set, the training set and verification set were randomly selected. When modeling in the whole growth period, the number of modeling set samples was 180 and the number of verification set samples was 100. However, when modeling in the single growth period, the number of samples in each period was 70, including 45 modeling set samples and 25 verification set samples, Training set and verification set are independent data. (Line 265-268 of revised manuscript).

  1. The spectral instrument described in this paper has a band range of 350nm-2500nm, but why is only 350-1800nm shown in Figure 2?

Response: We have added the relevant explanation. We used an ASD Field Spec Pro FRTM spectrometer (Malvern Panalytical, Malvern, UK) whose wavelength range is 350 – 2500 nm. However, as 1800nm-2500nm spectral information is mainly affected by environmental noise, water gas, it is difficult to invert the LAI spectral change rule. In this paper, only 350nm~1800nm spectral information is analyzed. (Line 154-157of revised manuscript).

  1. Vegetation index is constructed based on sensitive band. How to select the sensitive band of vegetation index in this paper? Please elaborate.

Response: Thanks for carefully reviewing the manuscript. We have added the relevant explanation. In this paper, we first analyzed the change rule of LAI and canopy spectrum in different growth periods of cotton under different applications of nitrogen and normalized, denoised, and smoothed the canopy spectrum data. The original spectrum was trans-formed into first-order differential spectra and the logarithm of the reciprocal spectra. The correlation between LAI and the original spectrum was analyzed along with the first-order differential spectrum and the logarithm of the reciprocal spectrum. The two bands that correlated the most closely were selected as the sensitive bands to establish the NDVI, RVI, EVI2, DVI and TVI and construct models to monitor the LAI in different growth stages of cotton and whole growth stages through models of regression analyses. This enabled us to explore the optimal vegetation index monitoring model of cotton LAI in different growth stages and whole growth stages to provide a theoretical basis to accurately monitor the changes in cotton LAI in various growth stages and suggest scientific fertilization. (Line 252-264 of revised manuscript).

Reviewer 3 Report

The study "Establishment of a Monitoring Model for the Cotton Leaf Area Index Based on the Canopy Reflectance Spectrum" is an excellent effort to monitor the cotton leaf area index.

The main conclusion of this study is that the use of sensitive bands for spectral reflectance allowed for the analysis of the correlation between cotton canopy spectral reflectance and LAI in different growth periods. This led to the development of a cotton LAI monitoring model, which was tested and found to be effective in determining optimal fertilization amounts and achieving real-time monitoring of agricultural conditions. Additionally, the study found that the sensitive bands for LAI vary in different growth stages of cotton, and that different vegetation index monitoring models are most effective in different stages of cotton growth.

The following questions should be answered before the paper can be accepted:

1. Can you provide more details on the experimental setup and methodology used to determine the correlation between cotton canopy spectral reflectance and LAI in different growth periods?

2. Can you provide more information on the different nitrogen treatments that were used in the study and how they affected the LAI of cotton in each growth stage?

3. Can you provide more details on the cotton LAI monitoring model that was developed and tested in this study, including the specific vegetation index models that were used?

4. Can you discuss the limitations and potential sources of error in the study, and how they may have impacted the results?

5. Can you provide more information on the potential applications of the findings of this study, including how the results can be used to optimize fertilization in cotton production?

6. Can you discuss how the results of this study compare to previous research on the use of spectral remote sensing for monitoring agricultural conditions, and how this study adds to the existing body of knowledge in this area?

Author Response

The study "Establishment of a Monitoring Model for the Cotton Leaf Area Index Based on the Canopy Reflectance Spectrum" is an excellent effort to monitor the cotton leaf area index.

The main conclusion of this study is that the use of sensitive bands for spectral reflectance allowed for the analysis of the correlation between cotton canopy spectral reflectance and LAI in different growth periods. This led to the development of a cotton LAI monitoring model, which was tested and found to be effective in determining optimal fertilization amounts and achieving real-time monitoring of agricultural conditions. Additionally, the study found that the sensitive bands for LAI vary in different growth stages of cotton, and that different vegetation index monitoring models are most effective in different stages of cotton growth.

The following questions should be answered before the paper can be accepted:

Response: We greatly appreciate the reviewer's comments. We have made the necessary improvements based on the suggestions.

  1. Can you provide more details on the experimental setup and methodology used to determine the correlation between cotton canopy spectral reflectance and LAI in different growth periods?

Response: Thank you for your advice. We have added the relevant explanation.

(1). Measurement of the canopy spectrum. The original sentence Measurements were performed in the bud stage (June 21), bloom stage (July 10), early stage(July 25), and full boll stage (August 18) of cotton growth using an ASD Field Spec Pro FRTM spectrometer (Malvern Analytical, Malvern, UK). The wavelength range of the spectrometer is 350–2500 nm; the resolution of the spectral region is 3 nm for 350–1000 nm and 10 nm for 1000–2500 nm. The spectral sampling interval is 1 nm, and the field of view is 25°. The measurements were performed in cloudless weather between 12:00 and 14:00 local time. The sampled plants had even growth and were free of diseases or pests. The instrument was calibrated using a whiteboard before the measurement. The probe of sensor was pointed downward, and the vertical distance from the probe to the cotton canopy was maintained at 50 cm. Ten spectral curves were collected for each sample point. The scanning time was set to 0.2 s, and the average of three spectral measurements was used.have been rewritten to“ We used an ASD Field Spec Pro FRTM spectrometer (Malvern Panalytical, Malvern, UK) whose wavelength range is 350 – 2500 nm. The resolution of the spectral region was 3 nm for 350 – 1000 nm and 10 nm for 1000 – 2500 nm. The spectral sampling interval was 1 nm, and the field of view was 25 °. When collecting spectra in the field, personnel should not wear white and particularly bright clothes to avoid reflection and affect spectral collection. The cotton canopy spectrum should be measured in the bud stage, flowering stage, early boll stage and full boll stage. The weather forecast should be studied in advance to select clear and cloudless weather for measurement and avoid cloudy and rainy weather. The time period of solar intensity in Xinjiang is 12:00 to 14:00, which is suitable to measure the spectrum. Plants that grew evenly without diseases and pests were selected for sampling. Before the measurement, the instrument and power supply were turned on, preheated for half an hour in advance, and then the whiteboard was used to calibrate the handheld computer that is connected to the data acquisition through Bluetooth. It was calibrated every 10 times to avoid the "drift" of the measurement results as time progresses, thus, re-ducing its accuracy. The sensor probe points downward, and the optical fiber probe is 50 cm away from the cotton canopy. Ten specification curves were collected at each sampling point. The scan time was set to 0.2 seconds, and data were collected as the average of the three spectral measurements. However, as 1800nm-2500nm spectral information is mainly affected by environmental noise, water and gas, it is difficult to invert the LAI spectral change rule. In this paper, only 350nm~1800nm spectral information is analyzed.

The spectrum acquisition device is as shown in the figure 1. It is primarily composed of a tripod, mobile tube, support plate, optical fiber, power supply, and handheld com-puter. First, the optical fiber was placed in the hollow mobile tube, and the optical fiber probe was fixed vertically downward. The mobile tube can move up and down on the tri-pod. It primarily adjusts the height of the optical fiber probe and the crop canopy. The canopy height in this experiment was 50 cm. The other end of the optical fiber is connect-ed to the power supply, which is fixed on the support plate of the tripod by nuts. The power supply should be turned down and preheated for half an hour in advance before taking measurements. It is then wirelessly connected through the handheld computer. The parameters are set; the data folders established, the whiteboard calibrated, and the spectra were collected each time. The sampling points differed each time. Because the leaves need to be collected later, the canopy structure is different, and secondary measurement cannot be conducted at the same sampling point. (Line 137-172 of revised manuscript)

Fig. 1.  The spectrum acquisition device

(2). Determination of the leaf area index. The original sentence Additionally, ground-based measurement of the cotton leaves was performed using the destruction method. After the spectral measurements were performed, samples were immediately removed from the measurement area. Three cotton plants were sampled from each sample point. The leaves were then removed, and each leaf was placed in a separate sealed box and immediately sent to the laboratory for analysis. The area of each leaf was measured using an LI-3100 desktop leaf area meter (LICOR, Lincoln, NE, USA). have been rewritten to “After the spectral measurement, ground based measurement of the cotton leaves was performed using the destruction method. We selected cotton with uniform growth, took two cotton plants at each sampling point, cut out all the leaves, unfolded the leaves, put them into a plastic bag, and recorded the number. Each cotton leaf was separately packed. To prevent the leaf from curling, put the plastic bag into the fresh-keeping box with ice box, The area of each leaf was measured using an LI-3100 desktop leaf area meter (LICOR, Lincoln, NE, USA). The sampling points differed each time. Because the leaves need to be cut off, the canopy structure is different, and secondary measurement cannot be conducted at the same sampling point. (Line 180-188 of revised manuscript).

  1. Can you provide more information on the different nitrogen treatments that were used in the study and how they affected the LAI of cotton in each growth stage?

Response: Thank you for your advice. We added the relevant explanation. Different rates of N application have obvious regulatory effects on the LAI of cotton during the whole growth period. Compared with other periods, cotton in the bud stage has less demand for nitrogen; the plants and LAI are small, and the plants photosynthesize weakly. There was little difference among the nitrogen treatments. At the flowering stage, the vegetative growth gradually shifted to reproductive growth, and the LAI and photosynthetic capacity increased. The LAI also increased in parallel with the nitrogen, but the N4 treatment started to decrease. Due to the vigorous growth of the plant, the leaves at the lower part of the plant were sheltered from each other, and there was poor ventilation. Thus, the leaves began to fall. At the boll stage, the LAI started to decrease as the growth period advanced. More nutrients were transferred from the leaves to the cotton bolls, which resulted in poor growth and little development of the leaves. They were small or yellow or even fall off. Therefore, the proper application of nitrogen can improve the photosynthetic capacity of leaves, prolong the time for efficient use of light energy, and make cotton reach the most appropriate growth state. The variation coefficient of cotton in the full boll stage was small overall, indicating that there was a stable variation coefficient of cotton LAI; the variation coefficient of bud stage was the largest, and there was a large difference in the cotton LAI. First, cotton growth needs to adapt to the local environment for a period of time during the early stage. Therefore, there is a large difference in plant growth under the influence of environmental factors, such as soil temperature and water. In the late stage, cotton has fully adapted to the local environment and can obtain nutrients to meet its own growth, and the difference in plant growth is reduced. (Lines 494-515 of the revised manuscript)

  1. Can you provide more details on the cotton LAI monitoring model that was developed and tested in this study, including the specific vegetation index models that were used?

Response: Thank you for carefully reviewing the manuscript. In this paper, we first analyzed the change rule of LAI and canopy spectrum in different growth periods of cotton under different applications of nitrogen and normalized, denoised, and smoothed the canopy spectrum data. The original spectrum was transformed into first-order differential spectra and the logarithm of the reciprocal spectra. The correlation between LAI and the original spectrum was analyzed along with the first-order differential spectrum and the logarithm of the reciprocal spectrum. The two bands that correlated the most closely were selected as the sensitive bands to establish the NDVI, RVI, EVI2, DVI and TVI and construct models to monitor the LAI in different growth stages of cotton and whole growth stages through models of regression analyses. This enabled us to explore the optimal vegetation index monitoring model of cotton LAI in different growth stages and whole growth stages to provide a theoretical basis to accurately monitor the changes in cotton LAI in various growth stages and suggest scientific fertilization. (Lines 252-264 of the revised manuscript)

  1. Can you discuss the limitations and potential sources of error in the study, and how they may have impacted the results?

Response: Thank you for your advice. We added the relevant explanation. This paper monitored the cotton LAI based on ground hyperspectral readings. There was high ground hyperspectral resolution highly accurate monitoring. Compared with the UAV and satellite remote sensing, the monitoring range was limited. It is difficult to obtain a wide range of crop growth conditions, and the readings need to be measured in a clear and cloudless condition. If they are collected in cloudy weather or in the morning and afternoon, the transmission, reflection, and absorption of light will be affected, and the collected spectral information will be highly inadequate. The reduction of sensitive spectral information makes it difficult to retrieve the vegetation parameter information, and it is difficult to find crops with insufficient nutrition and pests in time, which leads to the desiccation of plants or even death in the later stage. Therefore, in the later stage, the optimal vegetation index established by ground hyperspectral in each growth period can be used in UAV and satellite remote sensing. The UAV and satellite remote sensing data can be used to simulate the ground spectrum based on the spectral response function. Through cloud removal, noise removal, spectral transformation and other spectral preprocessing techniques and in-depth learning methods, the monitoring accuracy of UAV and satellite remote sensing can be improved, which also improves the ground hyperspectral and UAV complementary of the spectral information provided by satellite remote sensing. (Lines 625-642 of the revised manuscript)

  1. Can you provide more information on the potential applications of the findings of this study, including how the results can be used to optimize fertilization in cotton production?

Response: Thank you for carefully reviewing the manuscript. In this paper, different nitrogen treatments were designed for the four growth stages of cotton, including the bud stage, flowering stage, early boll stage and full boll stage. The cotton canopy spectrum and LAI were measured by ground hyperspectral readings, and the monitoring model of cotton LAI in each growth stage was established based on the vegetation index. We explored a model to monitor the optimal vegetation index that was applicable to different growth stages of cotton . The LAI can be estimated according to the optimal vegetation index model in different periods in the later stage. However, nitrogen has a significant regulatory effect on the LAI of cotton during the whole growth period. The level of plant nutrition can be clearly expressed through the leaves, and the LAI can effectively reflect the growth of the leaves, Thus, monitoring the LAI can indirectly reflect the crop nutrition status and provide further guidance for experts to conduct diagnose the levels of nutrition and recommend fertilization programs for different growth periods of cotton. (Lines 634-645 of the revised manuscript)

  1. Can you discuss how the results of this study compare to previous research on the use of spectral remote sensing for monitoring agricultural conditions, and how this study adds to the existing body of knowledge in this area?

Response: Thank you for carefully reviewing the manuscript. We added the relevant explanation, Most studies have applied the sensitive band and the established model to the monitoring of vegetation parameters during the whole growth period. Because cotton is in a dynamic growth process, the leaf structure, cell shape and nutrient content required in each period are different, and when the vegetation coverage reaches a certain range, it will not change. However, the LAI may still be increasing, so it is difficult to distinguish the change rule of LAI in different growth periods. The novelty of this study is that we designed different nitrogen treatments for cotton during the four growth stages of bud stage, flowering stage, early boll stage and full boll stage, and we then measured the cotton canopy spectrum and LAI using ground hyperspectral readings. The goal was to reveal the spectrum of cotton canopy and the change rule of the LAI under different nitrogen treatments in different growth stages of cotton and construct a model to monitor the LAI for each growth stage based on the vegetation index. We explored the optimal vegetation index monitoring model applicable to different growth periods of cotton to more accurately assess the LAI change rule of cotton in each growth period and improve the measures of managing cotton growth. (Lines 84-99 of the revised manuscript)
